# The Cytological Energy Detection of Purulent Inflammation in Synovial Fluid Is Not All Black and White

**DOI:** 10.3390/biomedicines12030667

**Published:** 2024-03-16

**Authors:** Petr Kelbich, Eliska Vanaskova, Karel Hrach, Jan Krejsek, Frantisek Smisko, Pavla Hruskova, Eva Hanuljakova, Tomas Novotny

**Affiliations:** 1Department of Biomedicine and Laboratory Diagnostics, Faculty of Health Studies, Jan Evangelista Purkinje University and Masaryk Hospital, 401 13 Usti nad Labem, Czech Republiceva.hanuljakova@kzcr.eu (E.H.); 2Department of Clinical Immunology and Allergology, Faculty of Medicine and University Hospital, Charles University in Prague, 500 03 Hradec Kralove, Czech Republic; eliska.vanaskova@kzcr.eu (E.V.); jan.krejsek@fnhk.cz (J.K.); pavla.hruskova@kzcr.eu (P.H.); 3Laboratory for Cerebrospinal Fluid, Neuroimmunology, Pathology and Special Diagnostics Topelex, 190 00 Prague, Czech Republic; 4Department of Orthopaedics, Faculty of Health Studies, Jan Evangelista Purkinje University and Masaryk Hospital, 401 13 Usti nad Labem, Czech Republic; tomas.novotny@kzcr.eu; 5Faculty of Health Studies, Jan Evangelista Purkinje University, 400 96 Usti nad Labem, Czech Republic; 6Department of Orthopaedics, Regional Hospital, 360 01 Karlovy Vary, Czech Republic; frantisek.smisko@kkn.cz; 7Department of Central Laboratories, Hospital Teplice, 415 29 Teplice, Czech Republic; 8Department of Orthopaedic Surgery, Faculty of Medicine and University Hospital, Charles University in Prague, 500 03 Hradec Kralove, Czech Republic

**Keywords:** synovial fluid, purulent inflammation, neutrophils, coefficient of energy balance, cytological energy analysis, purulent score

## Abstract

Neutrophils are frequently found in the cytological picture of synovial fluid in several joint pathologies, and a higher proportion of them can even wrongly indicate these cases as purulent inflammation. For reliable differentiation between purulent and non-purulent cases, we use the cytological energy analysis of the synovial fluid. Using this method, we examined 350 knee joint synovial fluid samples. Overall, we found that the percentage of neutrophils ranged between 20.0% and 50.0% in 44 (12.6%) cases and was above 50.0% in 231 (66.0%) cases. In the same group, only 85 (24.3%) highly anaerobic synovial fluid samples were evaluated as purulent inflammation, and another 17 (4.9%) cases were evaluated as very likely purulent inflammation. Further, we quantified the immediate risk of purulent inflammation using the “purulent score” (PS). Of the total of 350 samples, 103 (29.4%) cases were classified as having a very high risk of purulent inflammation (PS = 4), 53 (15.1%) cases were classified as having a significant risk of purulent inflammation (PS = 3), 17 (4.9%) cases were classified as having a moderate risk of purulent inflammation (PS = 2), and 75 (21.4%) cases were classified as having no immediate risk of purulent inflammation (PS = 1). Based on our results and analyses, the cytological energy analysis of synovial fluid is an effective method that can be used to detect and specify joint inflammation and the risk of septic arthritis development.

## 1. Introduction

The diagnosis of a septic joint infection is very serious; it is a life-threatening event that requires emergency diagnostics and therapy [1]. Septic joint arthritis can affect any age group; nevertheless, every group has its typical causative pathogen [2]. Risk factors are joints previously altered by degenerative changes or rheumatoid arthritis, patients older than 80 years, patients with previously surgically treated joints, immunocompromised patients, or patients with other septic joints. Medical history and blood serum marker analysis are important but nonspecific. Urgent laboratory diagnostics for septic arthritis are mostly based on the white blood cell count in the punctate and further isolation of the causative agent in synovial fluid and its possible subsequent analysis [3]. Immediate bacterial identification is necessary for targeted antibiotic therapy, except for urgent septic cases, where empirical antibiotic treatment should await diagnostic sampling [4]. The detection of a bacterial periprosthetic joint infection (PJI) can be ambiguous and differ from case to case. Due to the increasing number of arthroplasty implantations worldwide, with an expected PJI risk ratio, distinguishing between septic and aseptic synovitis seems relevant for our future clinical practice. Following the literature, the current diagnostic criteria for PJI are diverse and controversial [5]. There are various guidelines followed worldwide for the treatment of periprosthetic joint infection therapy [6,7,8]. In contrast to native joint septic arthritis, septic PJI can be chronic, with a lack of clinical signs like joint swelling. Synovial fluid analysis and agent analysis can be repetitively negative. The decision between PJI or mechanical complications of arthroplasty can then be assessed by the “experience factor” of the orthopedic surgeon. For both native joint arthritis and PJI, we have also successfully used a cytological energy analysis of the synovial fluid method in our department for a quick and reliable distinction between these two units [9,10,11,12,13,14].

### Theoretical Aspects of the Cytological Energy Analysis of Synovial Fluids

Neutrophils are dominant cells of innate immunity with numerous characteristics. The crucial strategy of neutrophils is an oxidative burst with the rapid release of reactive oxygen species (ROS). This leads to the killing of internalized microbes, especially extracellular bacteria and fungi [15,16,17,18,19,20,21].

The first reaction is the conversion of molecular oxygen into the superoxide anion (O_2_^−^). This reaction is catalyzed with the enzyme NADPH-oxidase, which plays a central role in the oxidative burst:2 O_2_ + NADPH → 2 O_2_^−^ + NADP^+^ + H^+^

The superoxide anion generates hydrogen peroxide (H_2_O_2_). This reaction is accelerated with the enzyme superoxide dismutase:O_2_^−^ + O_2_^−^ + 2 H^+^ → H_2_O_2_ + O_2_

The superoxide anion reacts with hydrogen peroxide to form a hydroxyl radical (OH⋅). This reaction is catalyzed with iron (Fe):O_2_^−^ + H_2_O_2_ → O_2_ + OH⋅ + OH^−^

The enzyme myeloperoxidase catalyzes the oxidation of chloride anion to hypochlorous acid (ClO^−^):H_2_O_2_ + Cl^−^ → ClO^−^ + H_2_O

These ROS damage the proteins, lipids, and DNA of the microbes and lead to their death. The oxidative burst of neutrophils is the basis of the purulent inflammatory response. Unfortunately, it is dangerous not only for microbes but also for the tissues of the host organism. Therefore, its persistence is undesirable. Generally, we consider the cytological energy analysis of extravascular body fluids as a useful direction for therapy to manage and prevent greater tissue damage during local inflammation.

Purulent inflammation in the joint is promptly observable in the synovial fluid. The oxidative burst consumes a large amount of oxygen and is accompanied by the intense development of anaerobic metabolism [19,22,23,24,25,26]. Therefore, it is necessary to evaluate the presence of neutrophils in the synovial fluid and their metabolic status. Many authors use glucose and lactate concentrations to assess this aspect; however, these methods have some limitations [27,28,29,30,31]. To illustrate, the concentration of glucose in the synovial fluid is affected by the concentration of glucose in the blood. Suppose, for example, that we follow glucose metabolism. In this case, we can see that the final lactate concentration is affected by the extent of anaerobic metabolism and the amount of glucose [32,33,34]. Therefore, we may prefer to assess the concentration of glucose with lactate using the coefficient of energy balance (KEB):$$\mathrm{KEB}=38-18\frac{\left[lactate\right]}{\left[glucose\right]}$$
Legend: [glucose] = molar concentration of glucose in the synovial fluid (mmol/L); [lactate] = molar concentration of lactate in the synovial fluid (mmol/L).

The coefficient of energy balance expresses the theoretical average number of adenosine triphosphate (ATP) molecules produced under appropriate metabolic conditions from one glucose molecule in the synovial fluid [9,10,12].

Normal conditions in the synovial fluid may be characterized by a predominance of aerobic metabolism with high ATP production, corresponding to high KEB values.

Activated immunocompetent cells have higher energy demands during inflammation [34,35]. Thus, these cells consume more glucose and oxygen, leading to the development of anaerobic metabolism and a decrease in ATP production [22,26]. A decrease in KEB values reflects these metabolic changes in the synovial fluid.

According to this, an energy pyramid may be constructed (Figure 1). The peak of this pyramid represents normal energy relationships (28.0 < KEB < 38.0). KEB values from 15.0 to 28.0 are characteristic of severe inflammation with increased energy requirements. KEB values below 10.0 represent strong anaerobic metabolism, typical of intense inflammation with an oxidative burst of neutrophils [9,10,12,24,26]. KEB values between 10.0 and 15.0 are evaluated as borderline.

At the same time, there is a large consumption of oxygen and intense development of anaerobic metabolism in neutrophils [15,24,26,36]. The presence of this inflammation in the joint is reliably reflected in the synovial fluid and can be followed using our cytological energy analysis.

The essence of detecting purulent inflammation in the synovial fluid with abundant neutrophils is to distinguish between aerobic metabolism and moderate or strong anaerobic metabolism [24,26]. Consistent with our previous studies, we consider KEB values > 15.0 as moderate anaerobic or aerobic metabolism and KEB values < 10.0 as strong anaerobic metabolism (Figure 2) [9,10,12].

Therefore, not every accumulation of neutrophils in the synovial fluid represents purulent inflammation. However, the abundant neutrophils must be considered. To assess the immediate risk, we therefore introduced the so-called purulent score (PS) into the examination scheme:

PS = 1: The absence of an immediate risk of purulent inflammation in the synovial fluid, with a low percentage of neutrophils (<20.0%).

PS = 2: A moderate risk of purulent inflammation in the synovial fluid in the case of a higher proportion of neutrophils (between 20.0% and 50.0%) with aerobic metabolism (KEB > 28.0).

PS = 3: A significant risk of purulent inflammation in the synovial fluid in the case of a high percentage of neutrophils (>50.0%) with aerobic metabolism (KEB > 28.0).

PS = 4: A high risk of purulent inflammation in the synovial fluid in the case of a significant proportion of neutrophils (>20.0%) with moderate anaerobic metabolism (15.0 < KEB < 28.0).

PS = 5: Purulent inflammation in the synovial fluid in the case of a significant proportion of neutrophils (>20.0%) and strong anaerobic metabolism (KEB < 10.0).

Figure 3 and Figure 4 present very similar cytological images of knee synovial fluid with a predominance of neutrophils. In the first case, it is non-purulent synovial fluid of the knee joint of a patient with juvenile idiopathic arthritis. The second case is purulent inflammation of the knee joint caused by *Staphylococcus epidermidis* infection.

## 2. Materials and Methods

This retrospective study was approved by the local Ethics Committee of Masaryk Hospital in Usti nad Labem, Czech Republic (reference number: 319/11). No informed consent was required for this study. The work did not involve any human experiments and did not require the collection of data outside of routinely taken parameters. All patient records and information were anonymized and deidentified before analysis.

We examined 350 synovial fluid samples from patients with knee joint disorders (Table 1). The main parameters evaluated were cytology, the molar concentration of glucose and lactate, the calculation of KEB value and the catalytic activity of aspartate aminotransferase (AST). We call this method “cytological energy analysis”.

### 2.1. Cytological Energy Analysis of Synovial Fluid

The synovial fluid samples were collected using test tubes without anticoagulants and immediately transported in accordance with ISO 15189 to our clinical laboratory [37]. We used an ultrasound-guided diagnostic puncture in the case of less-accessible joints.

The total number of elements was counted under an optical microscope using a Fuchs–Rosenthal chamber. The results were reported as the number of nucleated cells per 1 µL synovial fluid sample. Microscopic smears were prepared using the cytocentrifuge method immediately after receiving the sample in all cases. We used the cytospin StatSpin^®^ Cytospin CytoFuge^®^ (Beckman Coulter, Inc., Brea, CA, USA) in all cases. The speed of rotation was 1100 revolutions per minute and the rotation time was 10 min. Permanent cytological smears were stained using Hemacolor^®^ Rapid staining (Merck Co., Darmstadt, Germany). We applied the fixative solution, the eosin solution, and the azure staining solution for 20 to 30 s on every sample of synovial fluid. Finally, we used a mounting medium, Entellan (Merck Co., Darmstadt, Germany), in all cases. Microscopic analyses were performed by trained laboratory stuff using an Olympus BX40 microscope (Olympus, Tokyo, Japan) to determine the cellular composition of the synovial fluid. We used a magnification objective of 40× with a numerical aperture of 0.75 in all cases. The results were reported as percentages of neutrophils, eosinophils, lymphocytes, and monocytes.

Another portion of the samples was centrifuged using a centrifuge MPW-352 (MPW MED. Instruments, Katowice, Poland). The speed of rotation was 4500 revolutions per minute and the rotation time was 5 min. After centrifugation, the sample was analyzed on a Cobas 6000 analyzer (Roche Diagnostics, Rotkreuz, Switzerland).

The molar concentration of glucose was examined using the enzymatic reference method with hexokinase [38], as follows.

Hexokinase catalyzes the phosphorylation of glucose to glucose-6-phosphate via ATP:glucose + ATP → glucose-6-phosphate + ADP

Glucose-6-phosphate dehydrogenase oxidizes glucose-6-phosphate in the presence of NADP to gluconate-6-phosphate. The rate of NADPH formation during the reaction is directly proportional to the glucose concentration and is measured photometrically (340 nm):glucose-6-phosphate + NADP^+^ → gluconate-6-phosphate + NADPH + H^+^

The molar lactate concentration was measured using a colorimetric assay [39]:

L-lactate is oxidized to pyruvate by the lactate oxidase:L-lactate + O_2_ → pyruvate + H_2_O_2_

Peroxidase is used to generate a colored dye using the hydrogen peroxide generated in the first reaction:2 H_2_O_2_ + H donor + 4-aminoantipyrine → chromogen + 2 H_2_O

The intensity of the color formed is directly proportional to the L-lactate concentration. It is determined by measuring the increase in absorbance (700/660 nm).

The catalytic activities of aspartate aminotransferase (AST) were determined by using the IFCC method [40]:

AST in the sample catalyzes the transfer of an amino group between L-aspartate and 2-oxoglutarate to form oxaloacetate and L-glutamate:L-aspartate + 2-oxoglutarate → oxaloacetate + L-glutamate

The oxaloacetate then reacts with NADH, in the presence of malate dehydrogenase, to form NAD^+^:oxaloacetate + NADH + H^+^ → L-malate + NAD^+^

The rate of NADH oxidation is directly proportional to the catalytic AST activity. It is determined by measuring the decrease in absorbance (340 nm).

KEB values (KEB = 38 − 18 × lactate/glucose) were calculated for all samples, including rare cases with very low glucose concentrations below the measurement limit (=0.11 mmol/L). A glucose concentration of 0.11 mmol/L was then used as the minimum for all these deeply anaerobic cases.

### 2.2. Statistical Analysis

The values of nucleated cells, as well as AST, indicated high skewness and large differences in variability. Thus, instead of a parametric 2-way ANOVA, the nonparametric Scheier–Ray–Hare test was used to verify the effect of KEB and neutrophils on nucleated cell counts and AST values. The null hypotheses assume no effect of categorized KEB, neutrophils, or their interactions. A 5% significance level was considered.

## 3. Results

Neutrophils are very common immunocompetent cells, and they can be found in the synovial fluid of affected joints. Their predominance at all energy levels is also evident in the set of synovial fluid samples collected from the knee joints of our patients. This predominance increases with anaerobic metabolism (Figure 5).

Table 2 shows the distribution of synovial fluid examination results in our patient group according to the purulent score.

Of the total number of samples, 21.4% of cases had no immediate risk of purulent inflammation (PS = 1), 4.9% of cases had a moderate risk of purulent inflammation (PS = 2), 15.1% of cases had a significant risk of purulent inflammation (PS = 3), 29.5% of cases had a high risk of purulent inflammation (PS = 4), 4.9% of borderline cases had a high risk of purulent inflammation or purulent inflammation (PS = 4 to 5), and 24.3% of cases had purulent inflammation (PS = 5).

Figure 6 shows an increase in nucleated cell counts in the synovial fluid along with an increase in neutrophil frequencies and anaerobic metabolism presented by KEB values. These conclusions are supported by testing the influence of categorized factors (KEB and neutrophils) using the nonparametric Scheier–Ray–Hare test. There were significant effects of neutrophil frequencies (*p* = 0.015) and KEB values (*p* < 0.001), but no effects of their interaction (*p* = 0.287).

As shown in Figure 7, an increase in AST catalytic activities in synovial fluid accompanies an increase in anaerobic metabolism. In contrast, a direct relationship between the level of AST catalytic activity and the frequency of neutrophils is not apparent. These conclusions are supported by the result of testing the influence of categorized factors (KEB and neutrophils) using the nonparametric Scheier–Ray–Hare test. There is a significant effect of the KEB category (*p* < 0.001), but no effect of the categorized frequency of neutrophils Ngr (*p* = 0.339) or their interaction (*p* = 0.676).

## 4. Discussion

Different authors have used different cytological and biochemical parameters for the detection and specification of inflammation in different extravascular bodily fluids. The most common are nucleated cell counts and glucose and lactate concentrations. In our study, we used a so-called cytological energy analysis. Its essence is the inseparable assessment of the presence of immunocompetent cells and their metabolic activity by means of theoretical adenosine triphosphate production [9,10,11,12,13,14]. In the following, we compare the advantages of our method with the experience of some authors in detecting purulent inflammation in synovial fluid.

E. Pascual and V. Jovani considered the limitation of individual cases in terms of the diagnostic value of the percentage of polymorphonuclear leukocytes in the synovial fluid, which is in agreement with our opinion [29]. This limitation may be overcome by detecting the relevant immunocompetent cells and evaluating their metabolic activity in the synovial fluid. The lactate/glucose ratio was proposed by Berthoud et al. [41]. In our study, we determined the level of metabolic activity using the coefficient of energy balance (KEB) [9,10,12]. Both of these parameters provide identical information. In contrast with the lactate/glucose ratio itself, KEB follows the metabolic pathway and assists in the transformation of the concentrations of glucose and lactate in the synovial fluid into theoretical ATP production [21,32].

Berthoud et al. reported better results for the synovial lactate/glucose ratio than synovial lactate or glucose separately to differentiate septic arthritis from non-septic arthritis [41]. Our experiences are the same. The concentration of glucose in the synovial fluid is dependent on the level of glucose in the blood. Hypoglycemia can mimic a loss of glucose in the synovial fluid. However, hyperglycemia may lead to increased concentrations of glucose in the synovial fluid and mask its consumption, caused by the inflammation in the joint. Further, the scheme of glucose metabolism indicates that the concentration of lactate in the synovial fluid is not influenced exclusively by the extent of anaerobic metabolism in the synovial fluid, but also by the supply of the energy substrate [9,21,22,32]. Therefore, we performed a simultaneous assessment of both glucose and lactate using the lactate/glucose ratio or KEB calculation [9,10,12,41].

Berthoud et al. considered tissue hypoxia to be the cause of increased lactate concentrations. Further, they noted the anaerobic metabolism of synoviocytes as the reason behind decreased glucose concentrations in septic synovial fluid [41]. In our work, the interpretation of anaerobic metabolism is a consequence of the activation of immunocompetent cells during a local immune response in the synovial fluid. In particular, the oxidative burst of neutrophils in purulent inflammation consumes huge amounts of oxygen and therefore causes the rapid and intense development of anaerobic metabolism [21,22,24,26,32].

In our cohort of 350 patients, we found that 66.0% of cases featured a predominance of neutrophils, and 12.6% of cases had a significant presence of neutrophils ranging from 20.0 to 50.0% in the synovial fluid collected from the knee joint. We evaluated only 24.3% of these cases (with KEB < 10.0) as purulent inflammation and 4.9% (with KEB from 10.0 to 15.0) as highly suspicious of purulent inflammation. We categorized the neutrophil accumulation in the remaining 49.4% cases as preventive protection with an increased risk of damaging purulent inflammation. The five-step purulent inflammation scale (PS) then allowed us to effectively specify this risk level into no risk (PS = 1), moderate risk (PS = 2), significant risk (PS = 3), high risk (PS = 4), and purulent inflammation (PS = 5) [9,12].

Neutrophils are very efficient immunocompetent cells. However, their high activity with oxidative burst may also cause damage to the organism itself [42]. This phenomenon is evident in our patient cohort, in the sharp increase in AST catalytic activity with an increase in anaerobic metabolism (Figure 7) [12].

Negligible neutrophils (<20.0%) and mildly or moderately increased anaerobic metabolism (15.0 < KEB < 28.0) represent severe inflammation in the synovial fluid. The same cytological picture with strong anaerobic metabolism (KEB < 10.0) indicates intense inflammation with an oxidative burst of macrophages, as found in the synovial fluid from eight patients with a knee featuring total knee arthroplasty before reimplantation surgery and in one patient with gout synovitis [34].

## 5. Conclusions

Neutrophils are the dominant cells with innate immunity and are essential in the immune response. Nevertheless, their extreme activation with an oxidative burst associated with purulent inflammation may damage tissues. Therefore, the persistence of purulent inflammation is undesirable.

The oxidative burst of neutrophils consumes large amounts of oxygen very rapidly and leads to an intense increase in local anaerobic metabolism. This effect allows the reliable detection of purulent inflammation via cytological energy analysis. In this study, we used this method to diagnose an inflammatory response in the knee joints of 350 patients. The joint assessment of neutrophils and anaerobic metabolism in synovial fluid samples allowed us to recognize different degrees of knee joint compromise, from exclusion—through moderate, significant, and high risk—to complete purulent inflammation.

We consider cytological energy analysis to be a useful, commonly available, and inexpensive method that can be used to support the therapeutic management of joint inflammation.

## Figures and Tables

**Figure 1 biomedicines-12-00667-f001:**
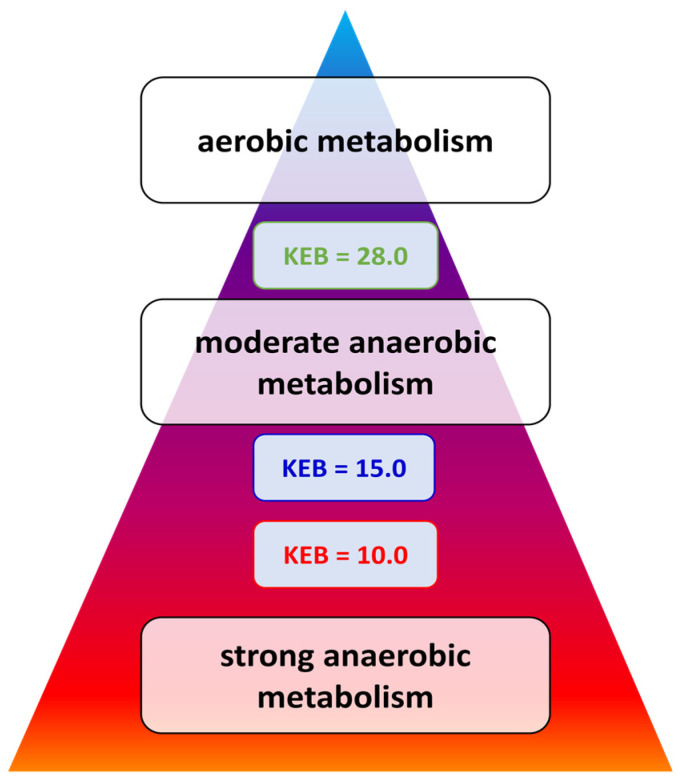
The energy pyramid.

**Figure 2 biomedicines-12-00667-f002:**
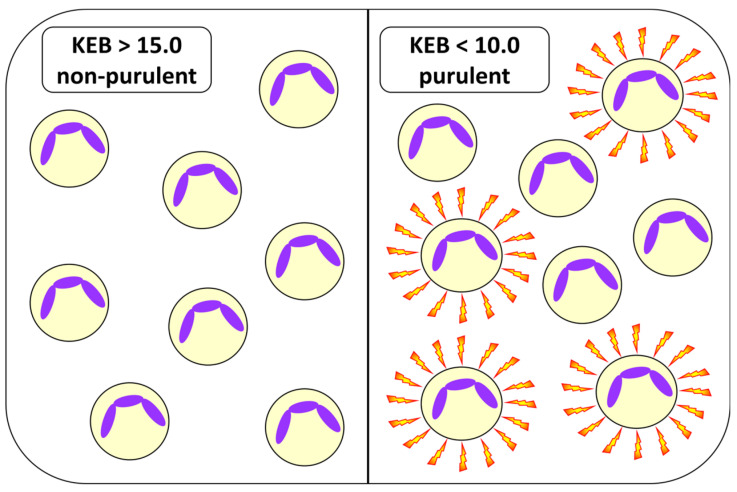
The accumulation of neutrophils without oxidative burst (non-purulent condition) and with oxidative burst (purulent inflammation).

**Figure 3 biomedicines-12-00667-f003:**
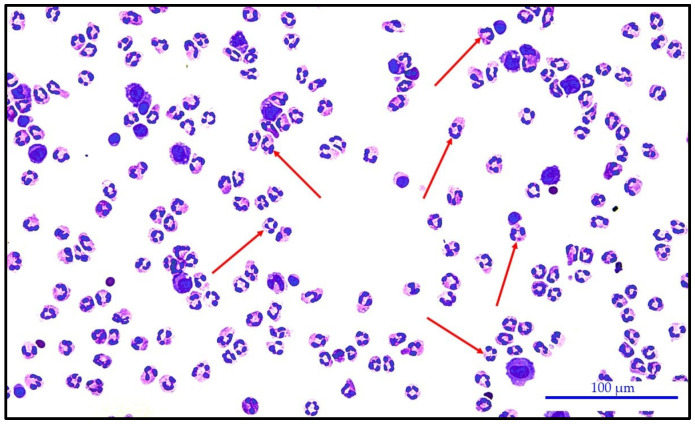
Predominance of neutrophils in the synovial fluid of a knee joint effusion of a patient with juvenile idiopathic arthritis. Neutrophils = 90.0%; KEB = 31.2; PS = 3 (red arrows point to several neutrophils). Leica DM2500 LED optical microscope with a Flexacam C3 camera (Leica Mikrosysteme Vertrieb GmbH, Wetzlar, Germany); HC PL FLUOTAR Lens 20×/0.55.

**Figure 4 biomedicines-12-00667-f004:**
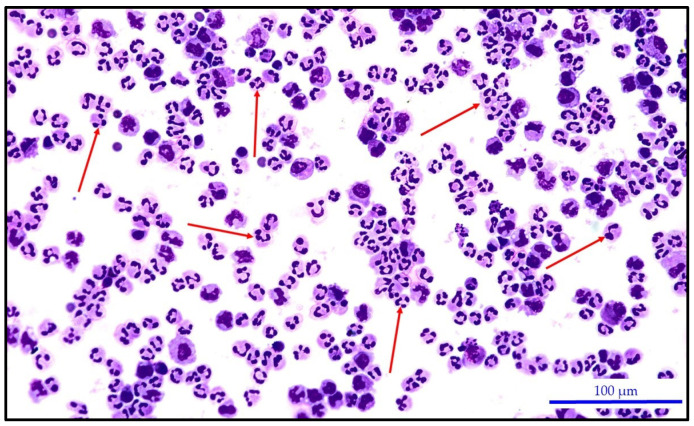
Predominance of neutrophils in the synovial fluid of a knee joint afflicted by purulent inflammation caused by *Staphylococcus epidermidis.* Neutrophils = 70.0%; KEB = −147.6; PS = 5 (red arrows point to several neutrophils). Leica DM2500 LED optical microscope with a Flexacam C3 camera (Leica Mikrosysteme Vertrieb GmbH, Wetzlar, Germany); HC PL FLUOTAR Lens 20×/0.55.

**Figure 5 biomedicines-12-00667-f005:**
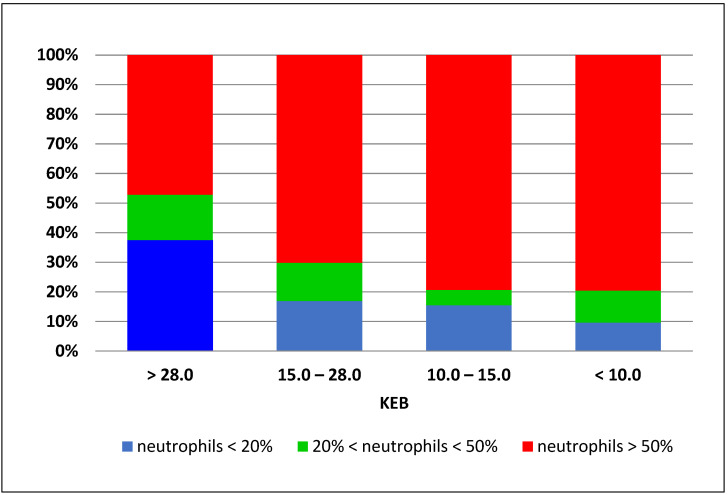
Distribution of the percentage of neutrophils according to the KEB values in the synovial fluids (*n* = 350).

**Figure 6 biomedicines-12-00667-f006:**
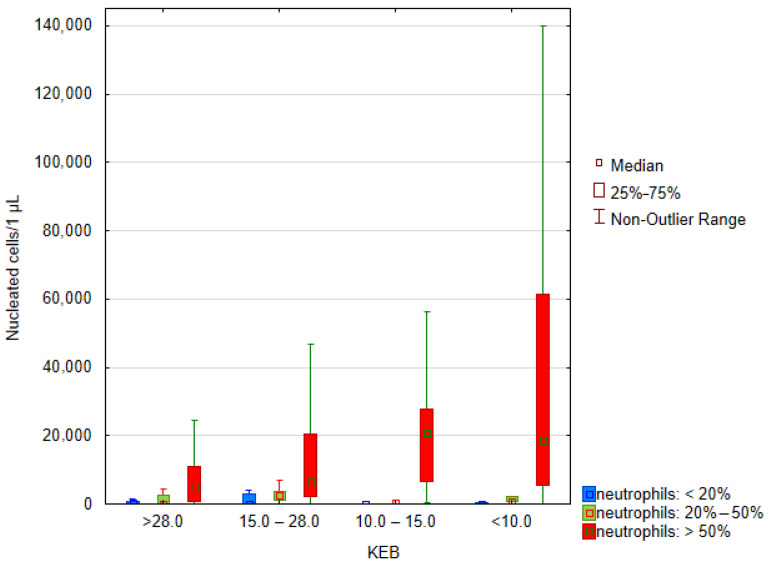
Boxplot of quantiles—nucleated cells in synovial fluid (*n* = 285, 65 outliers omitted), KEB categories, and neutrophil categories are statistically different.

**Figure 7 biomedicines-12-00667-f007:**
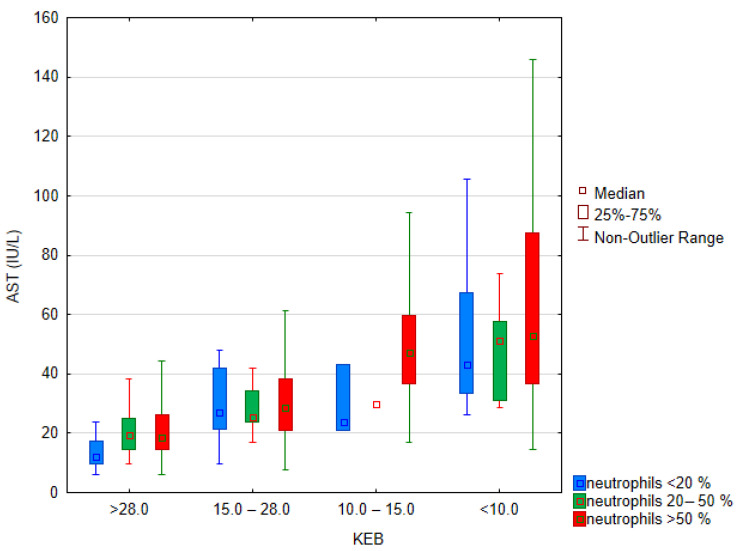
Boxplot of quantiles—catalytic activities of AST in synovial fluid (*n* = 332, 18 outliers omitted). KEB categories are statistically different.

**Table 1 biomedicines-12-00667-t001:** Demographic data of examined patients.

	Number	Median Age (Years)	Average Age (Years)	Minimum Age (Years)	Maximum Age (Years)
Female	132	55.5	48.2	3.0	96.0
Male	218	52.0	48.8	0.3	93.0

**Table 2 biomedicines-12-00667-t002:** Distribution of synovial fluids in accordance with the purulent score (PS).

KEB	Neutrophils < 20%	20% < Neutrophils < 50%	Neutrophils > 50%	Total
**>28.0**	*n* = 42 (12.0%)	*n* = 17 (4.9%)	*n* = 53 (15.1%)	*n* = 112 (32.0%)
	**PS = 1**	**PS = 2**	**PS = 3**	
**15.0–28.0**	*n* = 21 (6.0%)	*n* = 16 (4.6%)	*n* = 87 (24.9%)	*n* = 124 (35.4%)
	**PS = 1**	**PS = 4**	**PS = 4**	
**10.0–15.0**	*n* = 3 (0.9%)	*n* = 1 (0.3%)	*n* = 16 (4.6%)	*n* = 20 (5.7%)
	**PS = 1**	**PS = 4–5**	**PS = 4–5**	
**<10.0**	*n* = 9 (2.6%)	*n* = 10 (2.9%)	*n* = 75 (21.4%)	*n* = 94 (26.9%)
	**PS = 1**	**PS = 5**	**PS = 5**	
**total**	*n* = 75 (21.4%)	*n* = 44 (12.6%)	*n* = 231 (66.0%)	*n* = 350 (100.0%)

## Data Availability

All data used are with the author.

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
