# Peer review of "The Cytological Energy Detection of Purulent Inflammation in Synovial Fluid Is Not All Black and White"

_biomedicines, 2024, doi:10.3390/biomedicines12030667_

Round 1

Reviewer 1 Report

Comments and Suggestions for Authors

The topic is very relevant, the authors using the cytological-energy analysis of the synovial fluid as an effective method to detect and specify the inflammation in the knee joints.

The methodology is very modern and complex, using biochemistry, molecular biology, histology, immunohistochemistry and in vivo behaviour analysis.

The results have revealed that, calculating the purulent score allows to exclude purulent inflammation (PS = 1), to distinguish between moderate (PS = 2), significant (PS = 3) or high (PS = 4) risk of purulent inflammation and to detect full-blown purulent inflammation (PS = 5).

The conclusions are consistent with the evidence and arguments presented.

The references are very relevant, including also some relevant author’s previous experience in the field.

I suggest some minor editing corrections

1.       Line 91- is no need to give explanation In Czech, "Koeficient energetické bilance"

2.       In Fig 3, as I know, the final result of Krebs cycle is 36 ATP (not 38)

3.       Fig 6 should be bigger

4.       Line 226, mention the reference nr, after Berthoud et al.

Author Response

Dear Reviewer,

Firstly, on behalf of all the co-authors, we would like to thank the reviewer for taking their valuable time to elaborate on our manuscript. The comments of the reviewer were very apt and allowed us to optimize the structure of our work. We tried our best to meet the requirements in individual points and sincerely hope, that our manuscript is acceptable in its present form for publication in your highly esteemed journal. All revisions are indicated in red font in text (visible corrections form).

The topic is very relevant, the authors using the cytological-energy analysis of the synovial fluid as an effective method to detect and specify the inflammation in the knee joints.

The methodology is very modern and complex, using biochemistry, molecular biology, histology, immunohistochemistry and in vivo behaviour analysis.

The results have revealed that, calculating the purulent score allows to exclude purulent inflammation (PS = 1), to distinguish between moderate (PS = 2), significant (PS = 3) or high (PS = 4) risk of purulent inflammation and to detect full-blown purulent inflammation (PS = 5).

The conclusions are consistent with the evidence and arguments presented.

The references are very relevant, including also some relevant author’s previous experience in the field.

I suggest some minor editing corrections:

  1. Line 91- is no need to give explanation In Czech, "Koeficient energetické bilance"

Thank you for your comment. I deleted the text “Koeficient energetické balance”.

  1. In Fig 3, as I know, the final result of Krebs cycle is 36 ATP (not 38)

Yes, I agree. Many years ago, when I determined an equation for KEB, I accepted information about theoretical aerobic production of 38 ATP molecules from 1 molecule of glucose. Later I discovered information about theoretical production of 36 or fewer ATP molecules from 1 molecule of glucose. On the other hand, this fact doesn’t change the principle of KEB for evaluation of energy ratios.

  1. Fig 6 should be bigger

I have enlarged Figure 6.

  1. Line 226, mention the reference nr, after Berthoud et al.

I have added the citation number to the text.

Sincerely,

P. Kelbich et al.

Reviewer 2 Report

Comments and Suggestions for Authors

The paper is poorly written and very difficult to follow. Please, revise the manuscript before submitting it again.

Some examples:

Line 44-45 need to be revised.

There aren't enough citations to support the introduction's statements.

Figures 1  and 2 have not any scale bar. Not everyone is an expert in cytology. You should indicate the neutrophils in the photos.

Part 2 is materials and methods 

In the introduction, I have never seen the inclusion of figures in any MDPI journal before. They make the introduction very messy. Overall, the introduction needs to be represented after extensive revision.

How were the samples maintained during transportation? Did you ensure a certain temperature was monitored?

They cytological energy examination (paragraph n?)is very generic. Experiments must be reproducible and, at the moment, the tests are impossible to reproduce. Please, revise this part.

Underneath each figure, please write the number of repetitions, the statistical analysis.

The standard deviations in figure 10 are out of scale.

Line 228, what does it mean?

In the results, the syntax and the use of past/present tense must be revised.

Line 255, do not ask direct questions.

Line 258, what does "...than" mean?

Comments on the Quality of English Language

Extensive English revision. The writing is very difficult to understand.

Author Response

Dear Reviewer,

Firstly, on behalf of all the co-authors, we would like to thank the reviewer for taking their valuable time to elaborate on our manuscript. The comments of the reviewer were very apt and allowed us to optimize the structure of our work. We tried our best to meet the requirements in individual points and sincerely hope, that our manuscript is acceptable in its present form for publication in your highly esteemed journal. All revisions are indicated in red font in text (visible corrections form).

The paper is poorly written and very difficult to follow. Please, revise the manuscript before submitting it again.

Some examples:

  1. Line 44-45 need to be revised.

Thank you for your comment, we revised, reworded, and cited this sentence.

  1. There aren't enough citations to support the introduction's statements.

Thank you for your comment, we added 2 more citations to the introduction statement.

  1. Figures 1 and 2 have not any scale bar. Not everyone is an expert in cytology. You should indicate the neutrophils in the photos.

Thank you for your comment, I have added photos with scales and neutrophils marked.

  1. Part 2 is materials and methods 

Thank you for your comment, I have changed “Patients” to “Material”.

  1. In the introduction, I have never seen the inclusion of figures in any MDPI journal before. They make the introduction very messy. Overall, the introduction needs to be represented after extensive revision.

Thank you for your comment, I have revised the "Introduction", including the deletion of four figures.

  1. How were the samples maintained during transportation? Did you ensure a certain temperature was monitored?

Pre-analytical, analytical and post-analytical processes are carried out in accordance with ISO 15189, including monitoring of the temperature, and are strictly controlled by the Czech Accreditation Institute. All samples are therefore handled in accordance with ISO 15189. I have added this information to the “Materials and methods” section.

  1. They cytological energy examination (paragraph n?) is very generic. Experiments must be reproducible, and at the moment, the tests are impossible to reproduce. Please, revise this part.

I have described the cytological examination more precisely in the section "Materials and Methods".

  1. Underneath each figure, please write the number of repetitions, the statistical analysis.

The number of displayed values is commented underneath Figures 6 and 7.

  1. The standard deviations in figure 10 are out of scale.

Thank you for your comment. We assume it was Figure 6 - a new version of the figure was prepared with a larger y-scale (therefore some boxes are too small). Anyway, there are no SD values ​​in the quantile boxplot, the bars illustrate the non-outlier range.

  1. Line 228, what does it mean?

Thank you for your comment, I explained that the coefficient of energy balance was used instead lactate-glucose ratio for evaluation of metabolic activity in the synovial space.

  1. In the results, the syntax and the use of past/present tense must be revised.

Our manuscript underwent new language revision by the English Language Editing Services of MDPI.

  1. Line 258, what does "...than" mean?

We express the immunological importance of neutrophils together with their potential to damage tissues by the term "preventive protection with increased risk of purulent inflammation".

  1. Comments on the Quality of English Language

Our manuscript underwent new language revision by the English Language Editing Services of MDPI.

Sincerely,

P. Kelbich et al.

Reviewer 3 Report

Comments and Suggestions for Authors

1. Line 26 - grammatical error.

2. The research gap is not clear, therefore, the introduction section should be totally revised.

3. Figure 3 is not clear.

4. A consent form is still needed since the authors collected the samples from patients. The patients must agree with all the procedures.

5. Methods of the analysis are vague.

6. Discussions are not comprehensive and not comparing with other literature/findings.

7. Why not Anova test?

Comments on the Quality of English Language

Some grammatical errors found

Author Response

Dear Reviewer,

Firstly, on behalf of all the co-authors, we would like to thank the reviewer for taking their valuable time to elaborate on our manuscript. The comments of the reviewer were very apt and allowed us to optimize the structure of our work. We tried our best to meet the requirements in individual points and sincerely hope, that our manuscript is acceptable in its present form for publication in your highly esteemed journal. All revisions are indicated in red font in text (visible corrections form).

  1. Line 26 - grammatical error.

Our manuscript underwent new language revision by the English Language Editing Services of MDPI.

  1. The research gap is not clear, therefore, the introduction section should be totally revised.

Thank you for your comment, I've revised the "Introduction". I hope the current version is better and clearer.

  1. Figure 3 is not clear.

Thank you for your comment, I have deleted this figure.

  1. A consent form is still needed since the authors collected the samples from patients. The patients must agree with all the procedures.

Thank you for your comment. This is a retrospective study. We used only the results of routine examinations. This study was approved by the ethics committee of Masaryk Hospital in Usti nad Labem (reference number: 319/11). Informed consent was not required for this study. The work did not involve any human experiments and did not require data collection outside of routinely acquired parameters.

  1. Methods of the analysis are vague.

There was used only one type of statistical test, as mentioned in the “Statistical analysis” subsection of the Chapter 2. The rest of the statistical methods was simple description.

  1. Discussions are not comprehensive and not comparing with other literature/findings.

Thank you for your comment. We made an extensive revision of the discussion part. All revisions are indicated in red font in text (visible corrections form)

  1. Why not Anova test?

Explained in the “Statistical analysis” subsection of the Chapter 2: The values of nucleated cells as well as of AST performed high skewness and large differences of variability (see Figure 6, Figure 7). Thus, instead of parametric 2-way ANOVA, the non-parametric Scheier-Ray-Hare test was used.

  1. Comments on the Quality of English Language

Our manuscript underwent new language revision by the English Language Editing Services of MDPI.

Sincerely,

P. Kelbich et al.

Reviewer 4 Report

Comments and Suggestions for Authors

While the article is within the journal's scope, substantial revisions are necessary. The methodology requires referencing.

1.     How were the neutrophil grades graded? The method and reference should be mentioned.

2.     In the data analysis section, specify the considered significant levels.

3.     The significance or insignificance between groups in the chart should be determined with stars or letters.

4.     What biochemical, hematological, or sonographic findings can help validate the study results?

5.     Why was the AST liver enzyme used exclusively? Specify the method of measuring AST and the kit's manufacturer.

6.     Write about the limitations of the research.

7.       The conclusion should scientifically and explicitly present the study's results, determining clinical applications and research limitations. Direct use of results as presented in the results section of the article is not accurate.

8.     Writing revisions and language improvement should be made in the article.

Comments on the Quality of English Language

While the article falls within the journal's scope, it necessitates substantial revisions. I appreciate the esteemed editor for choosing me as a reviewer for the manuscript. Additionally, there is a need for improvement in the language used throughout the article.

Author Response

Dear Reviewer,

Firstly, on behalf of all the co-authors, we would like to thank the reviewer for taking their valuable time to elaborate on our manuscript. The comments of the reviewer were very apt and allowed us to optimize the structure of our work. We tried our best to meet the requirements in individual points and sincerely hope, that our manuscript is acceptable in its present form for publication in your highly esteemed journal. All revisions are indicated in red font in text (visible corrections form).

While the article is within the journal's scope, substantial revisions are necessary. The methodology requires referencing.

  1. How were the neutrophil grades graded? The method and reference should be mentioned.

Thank you for your comment. Optical microscopy was used to evaluate the cytological smears of synovial fluids. In all samples we counted the percentage of the main types of immunocompetent cells - neutrophils, eosinophils, lymphocytes and monocytes. I have described this procedure in the "Materials and Methods" section.

  1. In the data analysis section, specify the considered significant levels.

Thank you for your comment. A 5% significance level was considered - the last sentence in the „Statistical analysis“ subsection of the Chapter 2.

  1. The significance or insignificance between groups in the chart should be determined with stars or letters.

The significance is commented underneath Figure 6 and 7 now.

  1. What biochemical, hematological, or sonographic findings can help validate the study results?

Thank you for this comment. We added information about ultrasound guided diagnostic puncture we used in a case of a less accessible joints. We accept that joint infection diagnostics is a complex procedure. In all cases, we compared laboratory diagnostics results, clinical findings, sonography and x-ray, CT or MRI.

  1. Why was the AST liver enzyme used exclusively? Specify the method of measuring AST and the kit's manufacturer.

Thank you for your comment. When I started investigating extravascular body fluids many years ago, I discovered AST as an excellent parameter for assessing local tissue destruction. I therefore included this method as a fixed part of all routine examinations of extravascular body fluids, including cerebrospinal fluid, pleural effusions, abdominal effusions, synovial fluids, etc. (see below). I have described this method in more detail in the "Materials and Methods" section.

Kelbich, P.; Vachata, P.; Maly, V.; Novotny, T.; Spicka, J.; Matuchova, I.; Radovnicky, T.; Stanek, I.; Kubalik, J.; Karpjuk, O.; Smisko, F.; Hanuljakova, E.; Krejsek, J. Neutrophils in Extravascular Body Fluids: Cytological-Energy Analysis Enables Rapid, Reliable and Inexpensive Detection of Purulent Inflammation and Tissue Damage. Life 2022, 12, 160.

Matuchova, I.; Kelbich, P.; Kubalik, J.; Hanuljakova, E.; Stanek, I.; Maly, V.; Karpjuk, O.; Krejsek, J. Cytological-energy analysis of pleural effusions with predominance of neutrophils. Ther. Adv. Respir. Dis. 2020, 14, 1–10.

Kelbich, P.; Radovnický, T.; Selke-Krulichová, I.; Lodin, J.; Matuchová, I.; Sameš, M.; Procházka, J.; Krejsek, J.; Hanuljaková, E.; Hejčl, A. Can aspartate aminotransferase in the cerebrospinal fluid be a reliable predicitve parameter? Brain Sci. 2020, 10, 698.

  1. Write about the limitations of the research.

We have been using cytological-energy analysis of various extravascular body fluids for many years. We now present its results on a group of 350 patients with various knee joint disorders. This method is not suitable for all types of extravascular body fluids, such as urine and ejaculate.

  1. The conclusion should scientifically and explicitly present the study's results, determining clinical applications and research limitations. Direct use of results as presented in the results section of the article is not accurate.

Thank you for your comment. I have rewritten the "Conclusion" considering your comment.

  1. Writing revisions and language improvement should be made in the article. While the article falls within the journal's scope, it necessitates substantial revisions. I appreciate the esteemed editor for choosing me as a reviewer for the manuscript. Additionally, there is a need for improvement in the language used throughout the article.

Our manuscript underwent new language revision by the English Language Editing Services of MDPI.

Sincerely,

P. Kelbich et al.

Round 2

Reviewer 2 Report

Comments and Suggestions for Authors

Line 38-39, describe which are the current treatment of diagnosis.

How did you run the assay at line 176. If someone has to reproduce it, which are the steps?

Line 265, add citation

what does ''ensues'' mean?

Line 271, rephrase the sentence because it isn't clear what has a low sensitivity. The following sentence in unclear, too.

Comments on the Quality of English Language

Moderate English editing

Author Response

Dear Reviewer,

Thank you for your comments. In accordance with your requests, we have revised our manuscript as indicated below.

Sincerely,

Petr Kelbich et al.

Line 38-39, describe which are the current treatment of diagnosis.

This article focuses on the possibilities of periprosthetic and native joint infection diagnostics and the possibilities of distinguishing between septic and aseptic synovitis of the joint. The therapeutic options of this wide range of units exceed the aim of this article. 

Following your recommendation, we added the sentence in lines 39-40 and three new citations.

There are various guidelines followed worldwide for the treatment of periprosthetic joint infection therapy.

1) Osmon DR, Berbari EF, Berendt AR, et al. Diagnosis and management of prosthetic joint infection: clinical practice guidelines by the Infectious Diseases Society of America. Clin Infect Dis 2013; 56: e1–25.

2) Zimmerli W, Ochsner PE. Management of infection associated with prosthetic joints. Infection 2003; 31: 99–108.

3) Trampuz A, Zimmerli W. Prosthetic joint infections: update in diagnosis and treatment. Swiss Med Wkly 2005; 135: 243–251.

How did you run the assay at line 176. If someone has to reproduce it, which are the steps?

We have described the step between delivery of samples to our laboratory and their analysis using Cobas 6000 automatic analyzer.

Another portion of the samples was centrifuged using a centrifuge MPW-352 (MPW MED. Instruments, Poland). The speed of rotation was 4500 revolutions per minute and the rotation time was 5 minutes. After centrifugation the sample was analyzed on a Cobas 6000 automatic analyzer (Roche Diagnostics, Switzerland).

Line 265, add citation.

I added a citation.

what does ''ensues'' mean?

I changed „ensues“ to „follows“.

Line 271, rephrase the sentence because it isn't clear what has a low sensitivity. The following sentence in unclear, too.

We have rephrased the sentence and hope that now it is clearer.

Berthoud et al. reported better results for the synovial lactate/glucose ratio than synovial lactate or glucose separately to differentiate septic arthritis from non-septic arthritis. Our experiences are the same.

Moderate English editing

Our manuscript underwent a language revision by the English Language Editing Services of MDPI. Unfortunately, we are unable to improve our English further.

Reviewer 3 Report

Comments and Suggestions for Authors

All comments have been addressed.

Author Response

Dear Reviewer,

Thank you for your favourable review.

Sincerely,

Petr Kelbich et al.

Reviewer 4 Report

Comments and Suggestions for Authors

The requested revisions have been addressed. The article is now ready for publication consideration.

Author Response

(The authors gave the same response as above.)

Round 3

Reviewer 2 Report

Comments and Suggestions for Authors

The authors have replied all my comments.

Comments on the Quality of English Language

Minor.

Author Response

Dear Reviewer,

I would like to thank you for your precious time in reviewing our manuscript. Thanks to your comments, we were able to improve our manuscript significantly. 
Our manuscript underwent a language revision by the English Language Editing Services of MDPI. Unfortunately, we are unable to improve our English further.

Sincerely,

Petr Kelbich et al.